# Desmopressin for reversal of Antiplatelet drugs in Stroke due to Haemorrhage (DASH): protocol for a phase II double-blind randomised controlled feasibility trial

Michael J R Desborough [ORCID],[1] Rustam Al-Shahi Salman,[2] Simon J Stanworth,[3,4,5] Diane Havard,[6,7] Paul M Brennan,[2] Robert A Dineen,[6,7] Timothy J Coats,[8] Trish Hepburn,[9] Philip M Bath,[6,7] Nikola Sprigg[6,7]

► Prepublication history and supplemental material for this paper is available online. To view these files, please visit the journal online (http://dx.doi.org/10.1136/bmjopen-2020-037555).

For numbered affiliations see end of article.

**Correspondence to**
Dr Nikola Sprigg;
nikola.sprigg@nottingham.ac.uk

## ABSTRACT

**Introduction** Intracerebral haemorrhage (ICH) can be devastating and is a common cause of death and disability worldwide. Pre-ICH antiplatelet drug use is associated with a 27% relative increase in 1 month case fatality compared with patients not using antithrombotic drugs. We aim to assess the feasibility of conducting a randomised controlled testing the safety and efficacy of desmopressin for patients with antiplatelet-associated ICH.

**Methods and analysis** We aim to include 50 patients within 24 hours of spontaneous ICH onset, associated with oral antiplatelet drug(s) use in at least the preceding 7 days. Patients will be randomised (1:1) to receive intravenous desmopressin 20 μg in 50 mL sodium chloride 0.9% infused over 20 min or matching placebo. We will mask participants, relatives and outcome assessors to treatment allocation. Feasibility outcomes include proportion of patients approached being randomised, number of patients receiving allocated treatment, rate of recruitment and adherence to treatment and follow-up. Secondary outcomes include change in ICH volume at 24 hours; hyponatraemia at 24 hours, length of hospital stay, discharge destination, early death less than 28 days, death or dependency at day 90, death up to day 90, serious adverse events (including thromboembolic events) up to day 90; disability (Barthel index, day 90), quality of life (EuroQol 5D (EQ-5D), day 90), cognition (telephone mini-mental state examination day 90) and health economic assessment (EQ-5D).

**Ethics and dissemination** The Desmopressin for reversal of Antiplatelet drugs in Stroke due to Haemorrhage (DASH) trial received ethical approval from the East Midlands—Nottingham 2 research ethics committee (18/EM/0184). The DASH trial is funded by National Institute for Healthand Care Research RfPB grant: PB-PG-0816-20011. Trial results will be published in a peer reviewed academic journal and disseminated through academic conferences and through patient stroke support groups. Reporting will be in compliance with Consolidated Standards of Reporting Trials recommendations.

**Trial registration numbers** NCT03696121; ISRCTN67038373; EudraCT 2018-001904-12.

## Strengths and limitations of this study

► This is the first randomised controlled trial of desmopressin versus placebo for patients taking antiplatelet drugs with spontaneous intracerebral haemorrhage (ICH).
► Robust methodology developed from large international multicentre trials in spontaneous ICH such as TICH-2 and PATCH.
► Desmopressin and placebo are not identical so the person administering the intervention will not be masked to treatment allocation.
► All outcomes assessors are blinded to treatment allocation to minimise the risk of bias.

## INTRODUCTION

### Haemorrhagic stroke

Spontaneous intracerebral haemorrhage (ICH) causes approximately 3 million deaths per year worldwide[1] and is responsible for the loss of 64.5 million disability adjusted life years per year.[2] Two-thirds of survivors are left dependent on others.[3] Roughly one-third of patients are taking antiplatelet drugs at the time of ICH in high-income countries, and this proportion has been increasing over time.[4] Pre-ICH antiplatelet drug use is associated with a 27% relative increase in 1 month case fatality compared with patients not using antithrombotic drugs.[5] Despite development of effective treatments for ischaemic stroke (thrombolysis, thrombectomy, aspirin, hemicraniectomy) there is no proven effective drug treatment for ICH, although blood pressure lowering might improve outcome.[6]

### Haematoma expansion

Outcome after ICH is closely related to haematoma growth (expansion) which is

associated with worse outcome (death and disability).[7] Use of antiplatelet drugs or anticoagulants, time from onset of symptoms to baseline imaging, and ICH volume on baseline imaging are independent predictors of haematoma expansion.[8]

Platelet transfusion, tranexamic acid and recombinant activated factor VII have not been shown to improve outcomes after antiplatelet associated ICH, raising the need to consider alternatives.[9]

### Desmopressin

Desmopressin is a licensed pro-haemostatic drug that can be administered intravenously, subcutaneously, intranasally or orally and is used in a number of inherited bleeding conditions to treat or prevent bleeding.[10–12] Platelet function remains inhibited for 5–7 days after stopping antiplatelet drugs. Desmopressin stimulates release of Von Willebrand Factor (VWF) and factor VIII from endothelial Weibel-Palade bodies. VWF is responsible for platelet adhesion to collagen and may also bind platelets through their glycoprotein IIb/IIIa receptors,[13] so increased levels of VWF have the potential to compensate for the platelet function defect associated with antiplatelet drugs. Desmopressin may also increase the formation of procoagulant platelets.[14]

Desmopressin is recommended for the reversal of antiplatelet drugs in guidelines from the USA[15] but not in UK guidelines.[16] In a recent meta-analysis evaluating desmopressin for reversal of antiplatelet drugs for patients undergoing cardiac surgery, desmopressin was found to reduce blood loss, transfusion requirements and the need for a further operation due to bleeding.[17]

There are potential risks with use of desmopressin, including tachycardia hypotension, hyponatraemia and hyponatraemic seizures. A meta-analysis of 65 surgical randomised controlled trials, comparing desmopressin to placebo found no significant difference in rates of ischaemic stroke, myocardial infarction or venous thromboembolism.[18]

Desmopressin has been tested in ICH in four case series where it appears to be safe.[19–22] However, the series were small, non-randomised and did not have a placebo control arm, so it is not possible to make a clear assessment of the benefits and harms of administering desmopressin for these patients. American neurocritical care guidelines for the treatment of intracranial haemorrhage recommend consideration of 0.4 µg/kg desmopressin for treatment of antiplatelet-associated ICH,[15] although the level of evidence supporting this recommendation was recognised as very uncertain.

Searches of clinicaltrials.gov and WHO International Trials Clinical Registry Platform found no other ongoing randomised controlled trials of desmopressin vs placebo for reversal of antiplatelet drugs in stroke due to haemorrhage.

If it is feasible to recruit patients and to collect robust data then we intend to proceed to a definitive trial to test efficacy.

### Primary objective

To assess the feasibility of randomising, administering the intervention and completing follow-up for patients treated with desmopressin or placebo to inform a definitive trial.

## METHODS AND ANALYSIS

Desmopressin for reversal of Antiplatelet drugs in Stroke due to Haemorrhage (DASH) will be a multicentre double-blind randomised placebo-controlled, parallel group phase II feasibility trial. Participants will be enrolled by investigators from emergency departments or acute stroke units from 10 hospital sites in the UK. Patient flow though the trial is summarised in figure 1.

### Inclusion criteria

- ► Adults (≥18 years).
- ► Confirmed ICH on imaging.
- ► Randomised less than 24 hours from onset of symptoms (or from when last seen free of stroke symptoms).
- ► Prescribed and thought to be taking a daily oral antiplatelet drug in the preceding 7 days (cyclo-oxygenase inhibitors, phosphodiesterase inhibitors or P2Y$_{12}$ inhibitors).

### Exclusion criteria

- ► Known secondary causes of ICH (aneurysmal subarachnoid haemorrhage or haemorrhage due to transformation of infarction, thrombolytic drug, venous thrombosis, arteriovenous malformation or tumour).
- ► Patients at risk of fluid retention.
- ► Systolic blood pressure less than 90 mm Hg.
- ► Known drug eluting stent in previous 3 months.
- ► Allergy to desmopressin.
- ► Pregnant or breast feeding.
- ► Life-expectancy less than 4 hours or planned for palliative care only.
- ► Glasgow coma scale less than 5.
- ► Modified Rankin scale (mRS) more than 4.

### Consent

Patients with capacity to give consent will be approached directly. If this is not possible, a relative or close friend will be approached for consent, if this is not possible then a professional not associated with the trial will be approached for proxy consent. Patients will be approached again for consent when they regain capacity (online supplemental files 1–3; all trial documents are available at: http://dash-1.ac.uk/docs/public.php).

### Randomisation

Participants will be randomised centrally using a secure internet site in real-time. Treatment allocation will be concealed from all staff and patients involved in the trial. Randomisation involves minimisation on key prognostic risk factors: age (70 years or more); sex (male); time since onset (3 hours or more); systolic blood pressure (170 mm Hg or more); and presence (or no information on

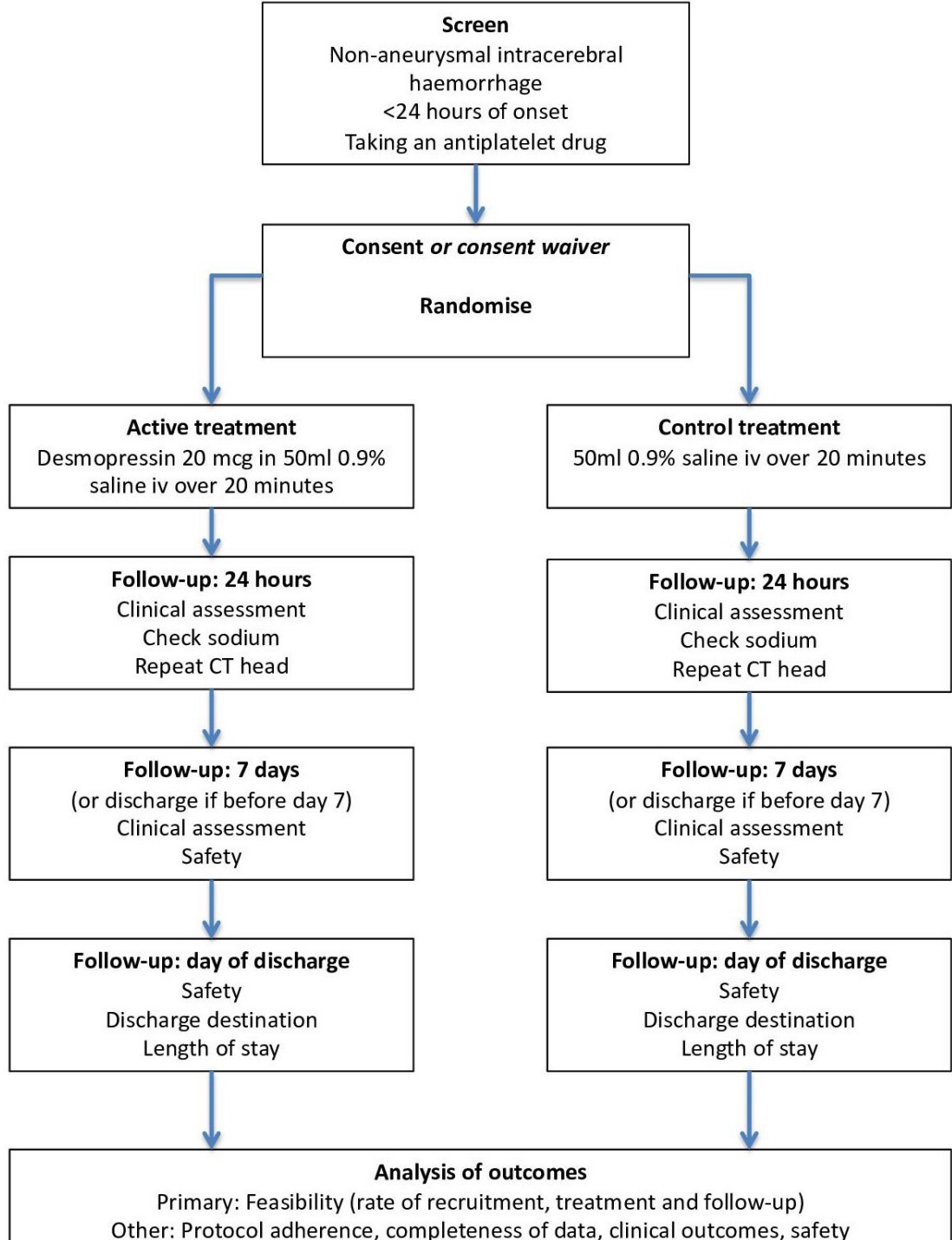

**Figure 1** Study flowchart.

presence or absence) of intraventricular haemorrhage. This approach ensures concealment of allocation, minimises differences in key baseline prognostic variables and slightly improves statistical power.[23] Randomisation will allocate a number corresponding to a treatment pack and the participant will receive treatment from the allocated numbered pack. It was considered not possible to mask/blind personnel preparing the injection for administration to the treatment allocation because the ampoules of active and placebo injections are different and carry the manufacturer's identifying information. Clinicians,

patients and outcome assessors (clinical, radiological and haematological assessors) will be blinded to treatment allocation.

In general, there should be no need to unblind the allocated treatment. If some contraindication to desmopressin develops after randomisation, the trial treatment should be stopped. Unblinding should be done only in those rare cases when the doctor believes that clinical management depends importantly on knowledge of whether the patient received desmopressin or placebo. In those few cases when urgent unblinding is considered

necessary, the doctor will call an emergency telephone number. The caller will then be told whether the patient received desmopressin or placebo. The rate of unblinding will be monitored and audited.

### Intervention

Patients will be randomised to either 20 μg intravenous desmopressin (DDAVP, 4 μg/mL 1 mL glass ampoules, Ferring Pharmaceuticals) or placebo of intravenous sodium chloride 0.9%. Placebo was selected for comparison because at present there are no established haemostatic therapies for ICH. Either desmopressin 20 μg (5×4 μg/mL) will be added to 50 mL sodium chloride 0.9% and infused over 20 min, or three 2 mL sodium chloride 0.9% will be added to 50 mL sodium chloride 0.9% and infused over 20 min. The investigational medicinal product (IMP) will be stored between 2°C and 8°C until use. All patients will receive all usual standard care including venous thromboembolism prophylaxis with intermittent pneumatic compression devices, blood pressure lowering therapy and neurosurgery if appropriate.

We will administer a single dose as the risk of continued bleeding and haematoma expansion after ICH is greatest in the first few hours. High levels of factor VIII and VWF are maintained for 6–8 hours after intravenous infusion therefore we believe a repeated dose at 12–24 hours would not be indicated. This dose regime (0.4 μg/kg body weight administered by intravenous infusion) is recommended in American neurocritical care guidelines for the treatment of intracranial haemorrhage,[15] although the level of evidence supporting this recommendation was recognised as very uncertain.

### Assessments

Participants' age, sex, medical history, antiplatelet drug use, ICH location, intraventricular haemorrhage and mRS will be assessed at baseline by local investigators. Participants will be reviewed again on day 2, hospital discharge and on day 90 to gather information on interventions, adverse events, discharge date and discharge destination. Data collection forms can be found in online (https://dash-1.ac.uk/).

At day 90 central assessors trained in the appropriate questionnaires and masked to treatment allocation will follow-up each patient by telephone. Barthel index, mRS, EuroQOL, cognition (mini-mental state examination) and serious adverse events will be assessed by telephone.

Brain imaging (CT head) will be undertaken as part of routine care before enrolment. A second research CT head scan will be performed 24 hours after treatment with the IMP to measure parenchymal haematoma volume. Haematoma volume will be measured centrally by trained assessors masked to treatment allocation using semi-automated segmentation. We will assess change in haematoma volume at 24 hours, and proportion of patients with haematoma expansion, defined as an absolute increase of greater than 6 mL or relative growth of greater than 33%.

To assess baseline platelet function, platelets will be stimulated with arachidonic acid (which is inhibited by aspirin) or adenosine diphosphate (which is inhibited by P2Y12 inhibitors). Platelet cell surface P-selectin expression (a measure of platelet activation) will then be measured using a standardised assay (Heptinstall; patent pending (PTC/GB2008/050169)) to assess retrospectively whether patients were taking antiplatelet drugs, as previously described.[24] VWF antigen, VWF activity and factor VIII (1-stage) will be measured centrally by assessors masked to treatment allocation. Blood samples will be drawn into 0.109M citrate and centrifuged at 2500g for 15 min. Platelet poor plasma will be aspirated and stored at −80°C at each site and then transferred to the central laboratory at the end of the trial for analysis. These tests will be performed for research only and the results will not be fed back to the clinical team. Serum sodium will be assessed locally 24 hours after administration of the IMP and these results will be available to the clinical team.

### Outcomes

This is a feasibility trial and the primary outcomes are based around feasibility: number of eligible patients who receive allocated treatment; rate of eligible patients randomised; proportion of eligible patients approached; proportion of eligible patients randomised and reasons for non-randomisation; adherence to intervention; proportion of participants followed up to 90 days and reasons for loss to follow-up; proportion of randomised participants with full outcome data available, and reasons for non-availability.

Secondary outcomes data will be collected to inform the design of a definitive trial but will not be statistically analysed. Secondary outcomes are change in ICH volume at 24 hours; hyponatraemia at 24 hours, length of hospital stay, discharge destination, early mortality less than 28 days, death or dependency at day 90, mortality up to day 90, serious adverse events (including thromboembolic events) up to day 90; disability (Barthel index, day 90), quality of life (EuroQol, day 90), cognition (telephone mini-mental state examination day 90) and health economic assessment (EQ-5D). Baseline platelet dysfunction will be measured and correlated with response to desmopressin; and change in factor VIII, VWF antigen and VWF activity will be assessed 1 hour after administration of desmopressin.

### Adverse events

All adverse events on day 1 (including during infusion) and for the 24 hours period post dose will be collected. All adverse events will be assessed for seriousness, expectedness and causality by adjudicators masked to treatment allocation. Serious adverse events will be categorised in accordance with the medical dictionary for regulatory authorities. As the IMP is administered once and has a short half-life, serious adverse events occurring within the first 7 days will be assessed for seriousness, expectedness and causality. In addition, fatal serious adverse events and

safety outcome events (fluid overload, hyponatraemia) will be reported until day 90.

## Statistical analysis

Data will be analysed by a qualified statistician who is blinded to treatment allocation, using a validated software package. A statistical analysis plan (SAP) will be agreed prior to database lock and release of randomisation codes. The trial will be reported in accordance with Consolidated Standards of Reporting Trials (CONSORT) guidelines including the extension to pilot and feasibility trials, as appropriate.[25 26] This is a feasibility trial and the main analysis will be with descriptive statistics only. Counts will be summarised using N and %, and continuous variables will be summarised using means and SD or medians and IQRs depending on their distribution. While some variables will be summarised by treatment group, no formal statistical comparisons will be made and any analyses will be considered purely exploratory. We will assess the feasibility of recruiting, treating and following up patients from 10 centres over 1 year. We will estimate a recruitment rate, treatment rate and follow-up rate. It is likely that a large definitive trial would be feasible if at least 50 participants were recruited into this study, that compliance with randomised treatment was high and that a high proportion of follow-up data were available. Lower recruitment would not preclude progression if there was some evidence that the barriers to recruitment identified could be overcome. A decision about feasibility for a larger trial will be made in conjunction with the Trial Steering Committee (TSC) taking into account recruitment rates, differences between centres and other relevant information.

This is a feasibility trial and as such will have no formal assessment of efficacy. The proposed primary efficacy outcome in a definitive trial would be death or dependency at day 90, measured using the mRS. Shifts in the mRS will be summarised for this trial but no formal confirmatory statistical analyses will be performed. Similarly, all other efficacy variables will be summarised using descriptive statistics. Serious adverse events will be summarised using descriptive statistics according to the treatment the participant received. Missing data will be reported. The investigation of this data and methods implemented to address the missing data, if appropriate, will be detailed in the SAP.

All available data will be used including overall numbers of patients presenting in clinic and screening data (where available). Where summaries by treatment group are provided, these will be based on an intention to treat population that is, according to the treatment the participant was randomised to, with the exception of safety data. A safety population will be defined to summarise the safety data in this study where participants will be summarised according to the treatment they received irrespective of randomisation. Summaries of the number and proportion of participants who would form a per protocol population in a larger trial, and reasons for exclusion from a per protocol population will be provided to allow future planning. No other data will be summarised for this population.

## Sample size and justification

Since this is a feasibility study with one of the objectives being to determine potential recruitment rates, a formal sample size calculation is not appropriate. If more than 50 participants were randomised from 10 centres in a 12-month period, it is likely that, assuming similar recruitment rates in additional centres, a larger study recruiting approximately 1200 participants in around 50 centres recruiting for 60 months would be feasible. Depending on the final sample size calculation for the definitive study, the number of centres and recruitment period could be determined using the information from the rates and patterns observed in the feasibility study. Information about set up times for centres will also better inform recruitment projections for a larger study. TICH-2 and other studies indicate approximately 25% of all people presenting with ICH are taking antiplatelet drugs.[27]

## Patient and public involvement

This study was developed in collaboration with, and is supported by Nottingham Stroke Research Partnership Group, made up of stroke survivors and carers. Members of the group have reviewed the proposed study, and in an iterative process, commented on its design and conduct. In particular, the group has influenced the approach to taking informed consent within the study. In addition, the proposal was also reviewed by the Research to Understand Stroke due to Haemorrhage patient reference group in Edinburgh (http://www.ed.ac.uk/clinical-brain-sciences/research/diagnoses-diseasetargets/rush/rush-patient-reference-group).

## Ethics and dissemination

The trial will be overseen by a TSC. A trial management committee based at the Stroke Trials Unit in Nottingham, UK will be responsible for day-to-day conduct of the trial. An independent Data Monitoring Committee (DMC) will review the data at 6 months. Study data will be collected, monitored and analysed in Nottingham. The data management policy is available at: http://dash-1.ac.uk/docs/public.php. The trial will be run in accordance with the principles of good clinical practice and the Declaration of Helsinki. The DASH trial has been granted ethics approval by the East Midlands—Nottingham 2 research ethics committee (18/EM/0184). The trial has been adopted in the UK by the National Institute for Health and Care Research.

Trial results will be published in a peer-reviewed academic journal. Reporting will be in compliance with CONSORT[25 26] recommendations. When the study is complete summary findings will be posted on the patient support group website. Findings will also be presented at conferences such as the UK Stroke Forum, European Stroke Conference, World Stroke Congress, British

Society for Haematology annual meeting and International Society on Thrombosis and Haemostasis annual meeting.

## Protocol amendments

Should a protocol amendment be made that requires ethics approval, the changes in the protocol will not be instituted until the amendment and revised informed consent forms and participant information sheets have been reviewed and received approval/favourable opinion from the research ethics committee and research and development departments. The results will be communicated to all principal investigators.

In the initial version of the protocol, which was in place from the start of the trial until 30 November 2019, participants could be recruited if they could be randomised less than 12 hours from onset of symptoms (or from when last seen free of stroke symptoms). In the updated version of the protocol, which came into effect on 01 December 2019, participants could be recruited if they could be randomised less than 24 hours from onset of symptoms (or from when last seen free of stroke symptoms).

## Confidentiality and access to data

All trial staff and investigators will endeavour to protect the rights of the trial's participants to privacy and informed consent, and will adhere to the Data Protection Act, 2018. The case report forms will only collect the minimum required information for the purposes of the trial. Case report forms will be held securely, in a locked room, or locked cupboard or cabinet. Access to the information will be limited to the trial staff and investigators and relevant regulatory authorities. Computer held data including the trial database will be held securely and password protected. All data will be stored on a secure dedicated web server. Access will be restricted by user identifiers and passwords. Information about the trial in the participant's medical records/hospital notes will be treated confidentially in the same way as all other confidential medical information. Electronic data will be backed up every 24 hours to both local and remote media in encrypted format.

## Insurance and indemnity

Insurance and indemnity for trial participants and trial staff is covered within the National Health Service (NHS) Indemnity Arrangements for clinical negligence claims in the NHS, issued under cover of HSG (96) 48. There are no special compensation arrangements, but trial participants may have recourse through the NHS complaints procedures. The University of Nottingham as research Sponsor indemnifies its staff, research participants and research protocols with both public liability insurance and clinical trials insurance. These policies include provision for indemnity in the event of a successful litigious claim for proven non-negligent harm.

## DISCUSSION

This will be the first randomised controlled trial comparing desmopressin to placebo for patients with ICH who are taking antiplatelet drugs. ICH is a common cause of death and disability worldwide. Patients taking antiplatelet drugs are more likely to have haematoma expansion and to have poorer outcomes than those who are not taking an antiplatelet drug. Early treatment with drugs to reduce the effect of antiplatelet drugs may reduce haematoma expansion. Desmopressin is a promising drug, which is commonly used in the treatment of inherited bleeding disorders and may reduce the effects of antiplatelet drugs.

In this trial we aim to determine if it is feasible to administer desmopressin within 24 hours to patients with ICH taking an antiplatelet drug. One potential barrier is that some patients will have delayed presentations to hospital. For those that reach hospital within 24 hours, we have aimed to make recruitment to the trial streamlined by minimising barriers to recruitment. This includes use of broad inclusion criteria and using a simple trial design. The design of this trial is based on the successful TICH-2[27] and PATCH[28] multicentre trials. The use of consent from a relative, close friend or professional representative is essential in this trial because the majority of patients will lack capacity.

If it proves feasible to run a trial of desmopressin compared with placebo for these patients then we aim to proceed to a large efficacy study. This will have an important impact on clinical practice. Desmopressin is widely available and inexpensive. It has the potential to reduce the risk of death and disability for patients with ICH who are taking antiplatelet drugs. Desmopressin could be rapidly adopted into clinical guidelines if it proves to reduce the risk of death or disability for patients with ICH.

## Trial sponsor contact

Ms Angela Shone, Head of Research Governance, Research and Innovation, University of Nottingham, East Atrium, Jubilee Conference Centre, Triumph Road, Nottingham, NG8 1DH.

## Trial Management Group

The Trial Management Group (TMG) will meet regularly, at least every 4 weeks to run the trial. TMG members are: Nikola Sprigg—Chief Investigator, Michael Desborough—Deputy Chief Investigator, Diane Havard—Senior Clinical Trials Manager, Sharon Ellender—Trial Coordinator, Lee Haywood—Programmer, Lisa Woodhouse—Trial statistician and Patricia Robinson—Trial Administrator.

## Trial Steering Committee

The independent TSC will provide oversight of the trial. It will meet (in person or by telephone conference) prior to commencement of the trial, and then at regular intervals

until completion (at least annually). Specific tasks of the TSC are:

► To approve the trial protocol.
► To approve necessary changes to the protocol based on considerations of feasibility and practicability.
► To receive reports from the Data Safety Monitoring Committee.
► To resolve problems brought to it by the coordinating centre and TMG.
► To ensure publication of the trial results.
► To advise on whether the main phase of the trial is feasible.

TSC members are: Colin Baigent—Chair, Christine Knott—patient and public involvement representative, Mathew Walters—patient and public involvement representative, Angela Shone—Sponsor, Trish Hepburn—Statistician, Philip Bath—Prof of Stroke Medicine, Rob Dineen—Prof of Neuro-radiology, Paul Brennan—neurosurgeon, Rustam Al-Shahi Salman—Prof of Stroke Medicine, Laura Green—Senior Clinical Lecturer in Transfusion Research & Innovation, Simon Stanworth—Associate Professor of Haematology, Tim Coates—Prof of Emergency Medicine, Phil Johnson—patient and public involvement representative, Emily Toon—Funder representative.

The trial coordinator, or where required, a nominated designee of the sponsor, shall carry out a site systems audit at least yearly and an audit report shall be made to the TSC.

## Data Monitoring Committee

The independent DMC will receive safety reports every 6 months, or more frequently if requested and perform unblinded reviews of safety data. The DMC will report their assessment to the independent chair of the TSC. Collaborators, and all others associated with the trial, may write through the trial office to the DMC, to draw attention to any concern they may have about the trial interventions, or any other relevant issues. There will not be an interim analysis. DMC members are: John Bamford—Chair, Graham Venables—Prof of Neurology, Martin Bland—Prof of Statistics.

## Participating UK centres

Nottingham University Hospitals NHS Trust; NHS Grampian; University College London Hospitals NHS Foundation Trust; NHS Lothian; Royal Devon & Exeter NHS Foundation Trust; University Hospitals of Leicester NHS Trust; University Hospitals of North Midlands NHS Trust; Derby Teaching Hospitals NHS Foundation Trust; Newcastle Upon Tyne Hospitals NHS Foundation Trust; St George's University Hospitals NHS Foundation Trust.

## Author affiliations
[1]Haemostasis and Thrombosis Centre, St Thomas' Hospital, London, UK
[2]Centre for Clinical Brain Sciences, University of Edinburgh, Edinburgh, UK
[3]Transfusion Medicine, NHS Blood and Transplant, Oxford, UK
[4]Department of Haematology, Oxford University Hospitals NHS Foundation Trust, Oxford, UK
[5]Radcliffe Department of Medicine, University of Oxford and NIHR Oxford Biomedical Research Centre, Oxford, UK
[6]Division of Clinical Neuroscience, University of Nottingham, Nottingham, UK
[7]Nottignham University Hospitals NHS Trust, Nottingham, UK
[8]Department of Cardiovascular Sciences, University of Leicester, Leicester, UK
[9]Clinical Trials Unit, University of Nottingham, Nottingham, UK

**Contributors** MJRD, SJS, RA-SS and NS conceived the ideas for the study. MJRD wrote the first draft of the protocol with RA-SS, SJS and NS. DH wrote the sections on trial regulation. TH wrote the statistical analysis plan. PMB, RAD, TJC and PMB critically reviewed the protocol and provided expert input.

**Funding** This paper presents independent research funded by the National Institute for Health Research (NIHR) under its Research for Patient Benefit (RfPB) Programme (Grant Reference Number PB-PG-0816-20011). The views expressed are those of the authors and not necessarily those of the NIHR or the Department of Health and Social Care. The trial sponsor and funders have no role in the; collection, management, analysis and interpretation of data; writing of the report; and the decision to submit the report for publication

**Competing interests** None declared.

**Patient consent for publication** Not required.

**Provenance and peer review** Not commissioned; externally peer reviewed.

**ORCID iD**
Michael J R Desborough http://orcid.org/0000-0002-1951-5616

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
