## [Reviewer comments · BMJ Open]

ARTICLE DETAILS

TITLE (PROVISIONAL)	Desmopressin for reversal of Antiplatelet drugs in Stroke due to Haemorrhage (DASH): protocol for a phase II double blind randomised controlled feasibility trial
AUTHORS	Desborough, Michael; Salman, Rustam; Stanworth, Simon; Havard, Diane; Brennan, Paul; Dineen, Rob; Coats, Timothy; Hepburn, Trish; Bath, Philip; Sprigg, Nikola

VERSION 1 – REVIEW

REVIEWER	Richard Lindley University of Sydney, Australia Like many in stroke research, I have collaborated extensively with international trialists. Therefore Philip Bath and I are both authors of the recent ENCHANTED trial publication in the Lancet (2019). I worked in the same department as Rustam Salman until 2003. I Chair a Data and Safety Monitoring Committee for a trial run by Nicki Sprigg.
REVIEW RETURNED	20-Mar-2020

GENERAL COMMENTS	This protocol is for a feasibility phase RCT testing desmopressin for spontaneous intracerebral haemorrhage - the DASH trial. Strengths of this manuscript include: the clarity of writing; the consumer co-design, including a proxy consent option for those who are acutely mentally incapacitated; feasibility outcomes; pre-specified notification of no statistical analysis of results given the small scale nature of the trial and feasibility outcomes; a sound rationale for the study; platelet function studies; imaging analysis as a potential biomarker of efficacy; and a team with an outstanding track record for acute stroke trials. Weaknesses are few but include the lack of double blinding (but this does not detract from the potential academic benefits of this study). I really had few suggestions but I don't like the abbreviation SICH given the common use of sICH to describe symptomatic intracranial haemorrhage in the context of the risks of thrombolysis for ischaemic stroke. I suggest you state that Spontaneous Intracerebral Haemorrhage stroke is abbreviated simply to ICH. The authors perhaps understate the global public health impact of ICH. Krishnamurthi et al (Lancet Glob Health 2013; 1: e259–81) demonstrated that although the absolute number of incident ischaemic strokes globally was twice that of haemorrhagic stroke, the overall global burden of haemorrhagic stroke (in terms of deaths and DALYs) was higher.
---

REVIEWER	Kimon Bekelis, MD The Dartmouth Institute for Health Policy and Clinical Practice
REVIEW RETURNED	21-Mar-2020

GENERAL COMMENTS	This is an important study in an area where we have very limited evidence. However, I am concerned that this study will be done in two steps with the first step being the feasibility study. If the authors want to utilize this step as preliminary data to secure robust funding and obtain traction then this is an appropriate step.
---

REVIEWER	J. David Spence Robarts Research Institute, Western University, London, ON, Canada
REVIEW RETURNED	20-May-2020

GENERAL COMMENTS	You say "Pre-stroke antiplatelet drug use is associated with a 27% relative increase in one-month case fatality compared to patients not using antithrombotic drugs." Presumably you mean in patients with intracerebral hemorrhage, but that is not clear. It is clear that this is a feasibility study. What is not clear is why you will be assessing many of the secondary outcomes in a study of only 50 patients. Some of them make sense, such as Baseline platelet dysfunction, correlation with response to desmopressin; and change in factor VIII, VWF antigen and VWF activity after administration of desmopressin. But why do length of hospital stay, discharge destination, early mortality less than 28 days, death or dependency at day 90, mortality up to day 90, serious adverse events (including thromboembolic events) up to day 90; disability (Barthel index, day 90), quality of life (EuroQol, day 90), cognition (telephone MMSE day 90), and health economic assessment (EQ-5D)? Are you intending to test your instruments and train study staff in the use of them?
--

VERSION 1 – AUTHOR RESPONSE

Reviewer: 1
Reviewer Name
Richard Lindley

Institution and Country
University of Sydney, Australia

Please state any competing interests or state 'None declared':
Like many in stroke research, I have collaborated extensively with international trialists. Therefore Philip Bath and I are both authors of the recent ENCHANTED trial publication in the Lancet (2019). I worked in the same department as Rustam Salman until 2003. I Chair a Data and Safety Monitoring Committee for a trial run by Nicki Sprigg.

Please leave your comments for the authors below
This protocol is for a feasibility phase RCT testing desmopressin for spontaneous intracerebral haemorrhage - the DASH trial.

Strengths of this manuscript include: the clarity of writing; the consumer co-design, including a proxy consent option for those who are acutely mentally incapacitated; feasibility outcomes; pre-specified notification of no statistical analysis of results given the small scale nature of the trial and feasibility outcomes; a sound rationale for the study; platelet function studies; imaging analysis as a potential biomarker of efficacy; and a team with an outstanding track record for acute stroke trials.

Weaknesses are few but include the lack of double blinding (but this does not detract from the potential academic benefits of this study).

I really had few suggestions but I don't like the abbreviation sICH given the common use of sICH to describe symptomatic intracranial haemorrhage in the context of the risks of thrombolysis for ischaemic stroke. I suggest you state that Spontaneous Intracerebral Haemorrhage stroke is abbreviated simply to ICH.

All sICH abbreviations changed to ICH as suggested

The authors perhaps understate the global public health impact of ICH. Krishnamurthi et al (Lancet Glob Health 2013; 1: e259–81) demonstrated that although the absolute number of incident ischaemic strokes globally was twice that of haemorrhagic stroke, the overall global burden of haemorrhagic stroke (in terms of deaths and DALYs) was higher.

We have added an additional reference and also the following sentence to the main text: “Spontaneous intracerebral haemorrhage (ICH) causes approximately 3 million deaths per year worldwide [1] and is responsible for the loss of 64.5 million disability adjusted life years per year [2].”

Reviewer: 2
Reviewer Name
Kimon Bekelis, MD

Institution and Country
The Dartmouth Institute for Health Policy and Clinical Practice

Please state any competing interests or state 'None declared':
None declared

Please leave your comments for the authors below
This is an important study in an area where we have very limited evidence. However, I am concerned that this study will be done in two steps with the first step being the feasibility study. If the authors want to utilize this step as preliminary data to secure robust funding and obtain traction then this is an appropriate step.

We have added an additional sentence to the end of the introduction to make this clear: “If it is feasible to recruit patients and to collect robust data then we intend to proceed to a definitive trial to test efficacy.”

Reviewer: 3
Reviewer Name
J. David Spence

Institution and Country
Robarts Research Institute, Western University, London, ON, Canada

Please state any competing interests or state 'None declared':
None declared

Please leave your comments for the authors below
You say “Pre-stroke antiplatelet drug use is associated with a 27% relative increase in one-month case fatality compared to patients not using antithrombotic drugs.” Presumably you mean in patients with intracerebral hemorrhage, but that is not clear.

Changed from “Pre-stroke” to “Pre-intracerebral haemorrhage”

It is clear that this is a feasibility study. What is not clear is why you will be assessing many of the secondary outcomes in a study of only 50 patients. Some of them make sense, such as Baseline platelet dysfunction, correlation with response to desmopressin; and change in factor VIII, VWF antigen and VWF activity after administration of desmopressin.

But why do length of hospital stay, discharge destination, early mortality less than 28 days, death or dependency at day 90, mortality up to day 90, serious adverse events (including thromboembolic events) up to day 90; disability (Barthel index, day 90), quality of life (EuroQol, day 90), cognition (telephone MMSE day 90), and health economic assessment (EQ-5D)? Are you intending to test your instruments and train study staff in the use of them?

We have added the following additional statement: “Secondary outcomes data will be collected to inform the design of a definitive trial but will not be statistically analysed.”

VERSION 2 – REVIEW

REVIEWER	Richard I Lindley University of Sydney, Australia
REVIEW RETURNED	26-Jun-2020
GENERAL COMMENTS	Thank you for addressing the referees comments. I had no further suggestions.